# Lung Imaging and Artificial Intelligence in ARDS

**DOI:** 10.3390/jcm13020305

**Published:** 2024-01-05

**Authors:** Davide Chiumello, Silvia Coppola, Giulia Catozzi, Fiammetta Danzo, Pierachille Santus, Dejan Radovanovic

**Affiliations:** 1Department of Health Sciences, University of Milan, 20122 Milan, Italy; 2Department of Anesthesia and Intensive Care, ASST Santi Paolo e Carlo, San Paolo University Hospital Milan, 20142 Milan, Italy; 3Coordinated Research Center on Respiratory Failure, University of Milan, 20122 Milan, Italy; 4Division of Respiratory Diseases, Luigi Sacco University Hospital, ASST Fatebenefratelli-Sacco, 20157 Milan, Italy; 5Department of Biomedical and Clinical Sciences, Università degli Studi di Milano, 20157 Milan, Italy

**Keywords:** artificial intelligence, lung imaging, CT, LUS, ARDS, COVID-19, deep learning, machine learning

## Abstract

Artificial intelligence (AI) can make intelligent decisions in a manner akin to that of the human mind. AI has the potential to improve clinical workflow, diagnosis, and prognosis, especially in radiology. Acute respiratory distress syndrome (ARDS) is a very diverse illness that is characterized by interstitial opacities, mostly in the dependent areas, decreased lung aeration with alveolar collapse, and inflammatory lung edema resulting in elevated lung weight. As a result, lung imaging is a crucial tool for evaluating the mechanical and morphological traits of ARDS patients. Compared to traditional chest radiography, sensitivity and specificity of lung computed tomography (CT) and ultrasound are higher. The state of the art in the application of AI is summarized in this narrative review which focuses on CT and ultrasound techniques in patients with ARDS. A total of eighteen items were retrieved. The primary goals of using AI for lung imaging were to evaluate the risk of developing ARDS, the measurement of alveolar recruitment, potential alternative diagnoses, and outcome. While the physician must still be present to guarantee a high standard of examination, AI could help the clinical team provide the best care possible.

## 1. Introduction

Broadly defined, artificial intelligence (AI) is a machine or computing platform that is capable of making intelligent decisions in a manner similar to the human mind [1]. In healthcare, AI could improve prognosis, diagnosis, treatment, and clinical workflow, particularly in the field of radiology, cardiovascular, and pathology [1,2]. Many of these medical tasks have been widely adopted in daily clinical practice [1]. During the last pandemic, significant progress was made in the development of AI resulting in more than 900 articles on COVID-19 and artificial intelligence [3]. Although the presence of a physician is still essential, AI could assist the clinical team in providing the best possible care.

During 2020, up to 30% of radiologic examinations were managed by AI with almost 20% planned for the following year [4,5], such as to detect intracranial hemorrhage, pulmonary embolism, and to monitor mammographic abnormalities. In addition, AI can improve scanning procedures, by reducing radiation exposure during scanning and acquisition, and then can optimize the sophisticated image reconstruction across magnetic resonance imaging, computed tomography (CT), and positron emission tomography modalities [4,5]. A recent survey found that radiologists would like to see AI improve anatomical measurements, lesion detection, and the quality of radiological imaging [4]. In particular, acute respiratory distress syndrome (ARDS) is a rather heterogeneous syndrome characterized by an inflammatory lung edema leading to an increased lung weight, decreased lung aeration with the presence of alveolar collapse, and interstitial opacities mainly in the dependent areas [6]. Lung imaging is an essential tool to assess not only the morphology but also the mechanical characteristics of ARDS patients. Lung CT and lung ultrasound (LUS) have a higher sensitivity and specificity than conventional chest radiography.

## 2. Machine Learning and Deep Learning

Machine learning (ML) is a subset of AI that focuses on developing algorithms and models that allow computers to learn from data and make predictions or decisions based on that data. These algorithms rely on manually engineered features. Developers write explicit instructions for the computer to follow. In ML, however, the computer learns patterns and relationships from data to make informed decisions or predictions. The primary goal of ML is to enable computers to improve their performance on a task over time by learning from examples rather than being explicitly programmed. ML algorithms are primarily designed to classify objects, detect patterns, predict outcomes, and make informed decisions [7].

In contrast, deep learning (DL) is a subfield of ML that focuses on training artificial neural networks with multiple layers (deep architectures) to learn complex patterns from data. In DL, models are based on deep artificial neural networks that can learn directly from data without the need for manual extraction. These neural networks are inspired by the structure and function of the human brain, where information is processed through interconnected neurons. The term “deep” in DL refers to the depth of the neural network, which consists of multiple hidden layers between the input and output layers. DL tends to work best with large datasets, as a large amount of data is required to successfully train deep neural networks. Certain neural network architectures such as the so-called convolutional neural networks (CNNs) are used specifically for image recognition. The potential applications of DL using lung CT and ultrasound may range from the early diagnosis, detection, and segmentation of specific lung regions to the prediction of the short- and long-term clinical outcomes [8].

## 3. Search Strategy

In this narrative review, we focus on the role of AI in the field of lung imaging in ARDS.

Figure 1 shows the search strategy flowchart.

We used PubMed and Embase databases performing two separate searches. For the first search, we used the following initial screening keywords: “lung CT scan AND artificial intelligence/machine learning AND ARDS/acute respiratory failure”. For the second search, we used “lung ultrasound AND artificial intelligence/machine learning AND ARDS/acute respiratory failure”. Then, the search was expanded using the following keywords: “lung imaging AND artificial intelligence/machine learning AND ARDS/acute respiratory failure”, resulting in 96 articles. After excluding duplicates, we also excluded screened abstracts and articles that did not include artificial intelligence involving either CT imaging or lung ultrasound imaging. Nineteen articles were identified. We excluded articles on pediatric patients and animals, resulting in 12 papers. We included 6 articles from the references. We included a total number of 18 studies in the review, which are summarized in Table 1.

## 4. Computed Tomography Scan

Lung CT scan has been used extensively for more than 20 years to improve our understanding of the pathophysiology of ARDS. In particular, the quantitative analysis of the ARDS lung CT scan has allowed the quantification of the amount of not aerated tissue, poorly aerated tissue, well-aerated tissue, and over inflated tissue, advancing the concept of the baby lung and of the lung as a “sponge model” [26]. This quantitative approach has shown the redistribution of the densities in prone position and the change in the not aerated tissue fraction at two airway pressures considered the gold standard for assessing recruitment in ARDS. In this context, CT scan has represented a useful tool to decide the better mechanical ventilation strategy [27]. Similarly, in the setting of chest trauma, the quantification of parenchyma damage, at hospital admission by CT scan, can help predict the evolution from the initial traumatic injuries to focal or diffuse alveolar hemorrhage followed by pulmonary edema and interstitial alterations typical of ARDS [28].

Then, recently during the COVID-19 pandemic, CT scan has played a relevant role as a screening tool due to its greater sensitivity for detecting early pneumonic changes. In fact, although COVID-19 is typically confirmed by viral nucleic acid detection, lung CT scan has been widely used to differentiate COVID-19 from other viral pneumoniae and to predict the severity of pneumonia even in the early stage [29].

The application of AI on CT scan lung images has been recently implemented in patients with COVID-19 disease to predict the evolution in ARDS, to stage and quantify the disease, and predict the outcome.

Similarly, in non-COVID-19 ARDS, AI has the potential to give a profound contribution, considering its ability to automatically and efficiently analyze and segment acutely injured lungs, to provide automated quantitative analysis, and to predict the development of ARDS, the alveolar recruitment, and the relationship between the quantitative analysis of lung tissue and specific outcomes.

### 4.1. Prediction of ARDS

The diagnosis and prediction of ARDS have been supported by various systems, tools, and techniques, both before and after the AI revolution. Imaging techniques such as chest radiography, CT, and LUS have played a critical role in the diagnosis and management of ARDS [30,31]. These lung imaging modalities have been essential in assessing lung aeration, predicting oxygenation response, and facilitating early diagnosis to prevent the progression of lung injury [32]. In addition, biomarkers have been explored for their potential in diagnosing ARDS and predicting its prognosis [33,34]. Recent advances in AI have significantly changed the landscape of traditional imaging techniques, allowing for more accurate and rapid analysis of medical imaging data for ARDS diagnosis and severity prediction [35].

AI-based diagnostic models combining clinical data and CT scans have been developed, providing accurate and explainable ARDS diagnostic models for real-life scenarios [36,37].

Recently, AI has facilitated the development of models to predict subsequent ARDS development using features identified at initial presentation with COVID-19, addressing the need for clinical decision support tools during the early stages of the COVID-19 pandemic [38,39].

In addition to COVID-19 pneumonia, blunt chest trauma is also currently associated with parenchymal lung injury to various extents, which may increase the risk of developing ARDS. Typically, the presence of an injury severity score (ISS) greater than 25 significantly increases the risk of developing ARDS [18]. Moreover, it has previously been shown that trauma patients with pulmonary contusions involving at least 20% of the total lung volume have a significantly higher risk of developing ARDS [40]. Thus, the ability to assess early information on lung CT that may be associated with the risk of developing ARDS could allow for timely supportive therapy. Röhrich et al. developed a ML method for the early prediction of ARDS based on CT in trauma patients at hospital admission. One hundred and twenty-three patients were enrolled. The model consisted of a fully automated ML and radiomics-based approach that showed a higher accuracy compared to an established score (ISS and abbreviated injury score of the thorax) to identify ARDS in trauma patients [18]. In this line, a rapid automated lung CT volumetry assessment of pulmonary contusions in trauma patients showed a good accuracy in assessing the risk of ARDS, length of intensive care stay, and time on mechanical ventilation [19].

### 4.2. Alveolar Recruitment

ARDS is characterized by widespread inflammation in the lung, leading to increased permeability of the alveolar–capillary barrier, impairment of pulmonary mechanical properties, and impaired gas exchange. There is often a phenomenon known as alveolar collapse or atelectasis, where some of the alveoli collapse and are not involved in gas exchange [41,42].

Mechanical ventilation itself can increase or cause lung damage known as ventilator-induced lung injury (VILI). Therefore, the therapeutic goal of mechanical ventilation in ARDS patients is not only to maintain “normal gas exchange” but also to protect the lung from VILI [43,44,45].

Lung protective strategies include low tidal volumes and adequate levels of positive end-expiratory pressure (PEEP) levels to keep the alveoli open and prevent them from collapsing. However, too high or too low levels of PEEP can lead to damage due to overdistension or cyclic collapse. Estimation of the percentage of lung that can be recruited or re-opened by applying transient increases in airway pressure has been demonstrated to be associated with the response to PEEP and prone position [27,46]. Thus, the quantification of alveolar recruitment can help the clinician optimize the protective ventilation strategy to avoid VILI.

The introduction of the lung CT quantitative analysis has allowed the assessment and quantification of the aerated and not aerated lung regions and the possible changes due to the mechanical ventilation and body position [6]. In particular, the application of the lung CT quantitative analysis performed at two different levels of airway pressure is considered the gold standard for the assessment of alveolar recruitment, because it can calculate the difference of not aerated tissue [47,48,49].

However, since the mid-1980’s, the application of quantitative analysis was rarely used in clinical practice because it requires the manual segmentation of the lung by the physicians [48]. The assessment of lung recruitment can take up to 6–8 h with a certain degree of error [15]. To improve the ability to assess lung recruitment, a visual anatomical evaluation of recruitment has been proposed [48].

Moving from the successful application of DL to the segmentation process of CT lung images in ARDS [50,51], using two CNNs architectures, the Seg-Net and the U-Net, Herrmann et al. decided to implement the U-net to develop a DL algorithm to automatically segment injured lungs affected by ARDS and to calculate lung recruitment by performing two CT scans at 5 and 45 cmH_2_O of airway pressure [11].

Training was performed on 15 healthy subjects (1302 slices), 100 ARDS patients (12,279 slices), and 20 COVID-19 patients (1817 slices): 80% of the patients were used for training and 20% for testing. The authors found that automatic lung segmentation performed by a properly trained neural network was reliable and closely matched the results obtained by manual segmentation. In fact, the total lung volume measured by AI and manual segmentation had a R^2^ of 0.99 and a bias of −9.8 mL (CI +56.0/−75.7 mL). Although the model was not perfect, especially in the most damaged lung areas, which are difficult to identify even for a trained radiologist, but which did not exceed 10% of the lung parenchyma, the AI segmentation showed the same degree of inaccuracy as the manual segmentation. In fact, for recruitability measured using manual and AI segmentation, change in not aerated tissue fraction had a bias of +0.3% (CI +6.2/−5.5%) while −0.5% (CI +2.3/−3.3%) was expressed for change in well-aerated tissue fraction.

Subsequently, Penarrubia et al. in a single center study assessed both intra- and interobserver smallest real difference exceeding measurement error of recruitment using both human and ML lung segmentation on CT scan [15]. Low-dose CT scans were acquired at 5 and 15 cm H_2_O of PEEP in 11 sedated and paralyzed ARDS patients and recruitment was computed as the change in weight of the not aerated lung regions. The intra-observer small real difference of recruitment was 3.5% of lung weight, while the human–human interobserver smallest real difference of recruitment was slightly higher amounting to 5.7% of lung weight, as also was the human–machine smallest real difference. Human–machine and human–human interobserver measurement errors were similar, suggesting that ML segmentation algorithms are a valid alternative to humans for quantifying alveolar recruitment on CT [15].

Furthermore, to overcome the difficulty in performing two CT scans at two different airway pressures, Pennati et al. developed a ML algorithm to predict lung recruitment in ARDS patients, starting from a single CT scan obtained at 5 cmH_2_O upon admission to the intensive care unit (ICU) [2].

The authors demonstrated that in 221 retrospectively analyzed ARDS patients, the use of four ML algorithms (logistic regression, support vector machine, random forest, XGboost) based on a lung CT scan at 5 cmH_2_O were able to classify lung recruiter patients with similar area under the curve (AUC) compared to a ML model based on the combination of lung mechanics, gas exchange, and CT data [2].

The application of this ML algorithm with an automatic lung segmentation and quantitative analysis could reduce the workload and ionizing radiation exposure of the traditional method of assessing lung recruitability.

### 4.3. Outcome

Concerning the outcome, hospital mortality has decreased over the decades, but has remained unchanged in recent years, despite advances in supportive care [52].

A small retrospective study of 42 patients with ARDS evaluated the relationship between the volume of well-aerated lung regions, calculated automatically by software, and outcome [9]. Total lung volumes and well-aerated lung regions were significantly higher in survivors. Estimates of the total volumetry and the regions of interest were obtained within three minutes with a very good reproducibility [9].

Several data have shown that lung volume and the amount of not aerated lung areas in COVID-19 are associated with respiratory severity and outcome [53]. Typical lung CT findings in COVID-19 patients include bilateral pulmonary ground-glass opacities and opacities with rounded edges usually localized in the peripheral lung regions [6].

Using a DL method to calculate the description of the CT, two clusters typically associated with COVID-19 and two clusters associated with bacterial pneumonia were found [12]. The clusters containing diffuse ground-glass opacities in the central and peripheral lung showed up to 91% accuracy in correctly classifying COVID-19 and pneumonia.

Liu et al. investigated the ability of quantitative lung CT analysis compared to traditional clinical biomarkers to predict progression to severe disease in the early stage of COVID-19 patients [14]. A group of 134 patients with COVID-19 who underwent lung CT scan and laboratory tests on day 0 and 4 were enrolled. All patients were followed up for 28 days until the first occurrence of severe disease or otherwise. Three AI-derived CT features were calculated according to Hounsfield units (−700/−500; −500/−200; and −200/−60 HU). The CT features at day 0 and day 4 and their changes from day 0 to day 4 showed the best discriminative ability to predict patient progression to severe disease. In this line, a retrospective study of COVID-19 patients used DL segmentation to assess lung volume and density composition [20]. The number of lung regions with a density between −549 and −450 of Hounsfield units was associated with an increased risk of ARDS. Although the results were not published, Lopes et al. proposed a multicenter retrospective longitudinal study to correlate the possible findings on lung CT in patients with COVID-19 infection and the course of the disease [16].

In COVID-19 patients, the use of the quantitative lung CT analysis at hospital admission, which calculates the volume of the affected lung as the sum of the poorly aerated and not aerated lung regions, predicted the need for oxygen support and intubation with good accuracy [13].

Regarding hospital mortality in COVID-19 ARDS based on AI quantification of lung involvement at hospital admission, the AI did not predict the outcome [17]. In contrast, the Sequential Organ Failure Assessment (SOFA) score resulted in an AUC for hospital mortality of 0.74 (95% CI 0.63–0.85), suggesting that other clinical parameters reflect the overall disease severity [54,55].

In severe COVID-19 ARDS patients with hypoxemia refractory to the conventional ventilation, veno-venous extracorporeal membrane oxygenation (ECMO) may be used to improve the outcome. However, the potential improvement in outcome is higher when ECMO support is applied in the early phase. Therefore, a possible early stratification should be considered. The use of an AI-based quantification of lung involvement was able to predict the need for ECMO with an acceptable AUC [0.83 (95% CI 0.73–0.94)] [10]. In addition, combining the SOFA score with CT lung involvement at ICU admission improved the AUC to 0.91 (95% CI 0.84–0.97) [10].

In summary, while the extent of lung involvement on imaging is an important consideration in assessing the severity of ARDS, there is not a specific “critical amount” that universally predicts outcome or guides the need for repeat imaging or ECMO use [10]. The decision to use ECMO is multifactorial and is based on clinical judgment, including considerations of the patient’s overall health, the underlying cause of ARDS, and the potential for recovery. Similarly, the timing and need for repeat imaging are individualized based on the clinical course of the patient and the judgment of the healthcare provider. In fact, the early clinical course of the disease may be more predictive of the outcome than the assessment at time of admission to the ICU [56].

## 5. Lung Ultrasound

LUS has been shown to be a useful tool in the assessment of numerous lung diseases and, in recent years, has proven to be also effective in the emergency care setting to screen patients with suspected COVID-19 pneumonia [57,58,59]. In fact, compared to traditional imaging, LUS has many advantages: it is radiation-free, inexpensive, rapid, bedside feasible, non-invasive, and lacks the laborious workflow of a CT scan. Considering all these features and its good accuracy as compared with lung CT scan, LUS is commonly used in the ICU to screen patients for ARDS [60,61,62,63,64]. Indeed, LUS has the potential to predict mortality in ARDS patients with a high level of accuracy (AUC 0.85). These findings also exhibit a strong correlation with the prognostic value derived from the invasively measured extravascular lung water index. Furthermore, in this condition, LUS is able to assess the likelihood of post-extubation distress after a successful spontaneous breathing trial, with an AUC of 0.86, and is also able to assess regional and global lung aeration [64,65,66,67]. LUS images in ARDS are characterized by the presence of a non-homogeneously distributed alveolar sonographic interstitial syndrome characterized by the presence of vertical artifacts (including the so-called “B lines” and the “white lung”), along with pleural thickening and consolidation in dependent regions [68,69]. However, these features, especially the vertical artifacts, are not specific for ARDS as they can be detected in many other pathological conditions (i.e., pulmonary edema, pneumonia, and pulmonary fibrosis). In addition, LUS interpretation can be limited by operator confidence in image acquisition and interpretation, which can lead to intra-reader variability and a limited inter-reader agreement [70,71].

To overcome these limitations and to curb operator-related variability, AI has recently been employed in different medical areas to aid LUS image analysis and interpretation [14,72], such as emergency and intensive care settings [3,73].

DL has the ability to directly process and gather intermediate and advanced features obtained from raw data, such as ultrasound images, and then make intelligent decisions based on the learned features. The absence of cognitive bias or the need for spatial pixel connections allows DL to treat images as numerical sequences, enabling the evaluation of quantitative patterns that could unveil insights beyond human interpretation thereby enhancing human diagnostic capability. According to the type of the skill requested (i.e., classification, detection, and segmentation), there are mainly three types of DL architecture: supervised deep networks or deep discriminative models, unsupervised deep networks or deep generative models, and hybrid deep networks. Supervised deep networks are the most widely used in ultrasound imaging, the major methodology of interest being the CNN [14,74,75].

Few studies are currently available regarding the use of DL in LUS for the evaluation of ARDS in non-COVID-19 patients. During the recent COVID-19 pandemic, LUS disease-specific patterns showed a higher sensitivity compared to chest X-ray in the identification of COVID-19 pneumonia [76,77], making this disease model the predominant focus of DL application. Indeed, the automated assessment enabled by DL ensures a prompt diagnosis in situations where resources and trained personnel are scarce, ideally addressing such challenges.

### 5.1. Prediction of ARDS Diagnosis

Two studies investigated the possibility of introducing DL modalities to discriminate different stages of parenchymal changes secondary to pneumonia [21] by grading vertical artifacts [22] of LUS.

Baloescu et al. designed a new custom DL that operated on dynamic ultrasound data for automated assessment of sonographic lung B lines. The DL consisted of a CNN developed using 2415 sub-clips of 12 frames each from 400 emergency department patients. Each sub-clip was evaluated by two emergency physicians with expertise in LUS, using a predeterminate ordinal scale from 0 (none) to 4 (severe). In addition, a binary classification was performed pooling together as “normal” the images with score 0 or 1 and as “abnormal” the images with score 2–4. The experts’ rating was used as ground truth and compared with the interpretations given by the new DL model using 100 sub-clips not used during the DL training. Considering the assessment of presence/absence of B lines, the new DL model showed an overall accuracy of 94% with kappa of 0.88; however, for the severity assessment, the overall accuracy was only 56% with kappa of 0.65, showing that the new algorithm is better at distinguishing B lines but not their severity [22].

Zhang et al. investigated the feasibility of computer-assisted ultrasound diagnosis using three CNN-based DL models—VGG, ResNet, and EfficientNet—for the detection and classification of pneumonia based on a self-made LUS image dataset built on a total of 10,350 LUS images. Each image of the dataset was manually classified into eight clinical features of pneumonia (0 = normal; 1 = B lines < 3; 2 = B lines > 3; 3 = area of merging B line is less than half; 4 = area of merging B line is more than half; 5 = depth of pieces is less than 1 cm; 6 = air bronchogram and depth of parenchymal hepatization is less than 3 cm; 7 = pleural effusion and depth of parenchymal hepatization is more than 3 cm). Since for some of the features evaluated by Baloescu there were not enough images for training and testing sets, several clinical features were manually grouped together into different “classes” resulting in three different datasets: one including three classes (class 1: feature 0; class 2: features 1–4; class 3: features 5–7), one including four classes (class 1: feature 0; class 2: features 1–4; class 3: features 5–6; class 4: feature 7), and the last one encompassing eight classes, i.e., a class for each of the eight features. All of the three datasets were compared across classification models and the EfficientNet showed to be the best model providing for the three and four classes datasets an accuracy of 94.62% and 91.18%, respectively, whilst the best classification accuracy of the eight classes dataset was only 82.75% [21].

### 5.2. Differential Diagnosis

AI has been applied to LUS imaging for its potential role in differentiating healthy subjects from COVID-19 pneumonia and ARDS, hydrostatic pulmonary edema, and bacterial pneumonia and ARDS.

Born et al. proposed another DL LUS model able to distinguish COVID-19 from healthy subjects and bacterial pneumonia with a sensitivity of 0.90 ± 0.08 and a specificity of 0.96 ± 0.04. The model was developed using a dataset made by 261 recordings from a total of 216 patients affected with COVID-19, bacterial pneumonia, non-COVID-19 viral pneumonia, and healthy controls. Due to data availability, the three non-COVID-19 viral pneumonia videos were excluded. Five DL models were then compared in terms of recall, precision, specificity, and F1 scores. Overall, both VGG and VGG-CAM showed encouraging results, achieving an accuracy of 88 ± 5% in the detection of COVID-19 pneumonia, across a 5-fold cross-validation with 3234 frames [23]. Arntfield et al. developed another CNN able to discriminate between similar appearing LUS images with pathological B lines of three different origins (COVID-19 ARDS, non-COVID ARDS, and hydrostatic pulmonary edema) using a total of 612 LUS videos from 243 patients (84 COVID-19 ARDS, 78 non-COVID-19 ARDS, and 81 hydrostatic pulmonary edema). To assess the CNN performance, a subset of 10% of the total data was used, not previously used during the training process. The evaluation made by CNN was then compared to the LUS interpretation given by experienced physicians completing an online interpretation exercise. The trained CNN performance on the independent dataset showed an ability to discriminate between COVID-19 (AUC 1.0), non-COVID-19 ARDS (AUC 0.934), and pulmonary edema (AUC 1.0) pathologies. This was significantly better than the physicians’ ability (AUCs 0.697, 0.704, and 0.967 for the COVID-19 ARDS, non-COVID-19 ARDS, and pulmonary edema classes, respectively; *p* < 0.01), showing that a trained neural network is able to detect subvisible features within LUS images [24].

Ebadi et al. proposed a fast and reliable DL model, specifically the Kinetics-I3D network, using LUS scans to explore the possibility of detecting and differentiating ARDS from pneumonia. Compared to other DL models, this trained model was able to classify an entire LUS scan obtained at the point-of-care, eliminating the need for preprocessing or analyzing frames individually, since the neural network could be retrained with new data to adapt the model to the needs of specific LUS applications. The results obtained with the new DL methods were benchmarked against ground truth assessed by expert radiologists showing an accuracy of 90% and a precision score of 95%. Moreover, the proposed model was very rapid as it was able to process the entire scan with a single forward pass into the network, avoiding time-consuming frame-by-frame analysis [25].

### 5.3. Limitations of AI in LUS

To date, the application of DL in thoracic echography has been very limited as compared to other imaging techniques. One of the reasons is the limited availability of organized LUS databases. In fact, to reach an optimal learning performance, a wide number of labeled LUS images is needed. This requirement can be challenging as LUS is an evolving technique, and currently, there are only a limited number of experts capable of providing a suitable interpretation.

Indeed, to date, LUS training for ARDS has often been the prerogative of emergency department and ICU staff, lacking the structured, shared, and formal reporting typical of other radiologic tests such as lung CT, which may limit standardization and uniform informative input for DL. On the other hand, the majority of the radiology training programs do not include education in LUS interpretation.

The prevalent issue arising from a deep model with limited training samples is overfitting that can be addressed by two different approaches: model optimization and transfer learning. Model optimization focuses on making the DL model itself to work better with available data using different types of strategies (e.g., well-designed initialization/momentum strategies, efficient activation functions, dropout, and batch normalization, stack/denoising), whereas transfer learning utilizes knowledge from one domain to enhance the performance in another domain with limited data.

Another limitation is that many of the shared LUS databases lack a complete interpretation of the thorax (since the evaluations are mostly performed with a focused approach) and important information such as patient details and technical or setting data. Collecting these data should help differentiate similar LUS patterns that are only apparently non-disease specific, such as those observed in patients with ARDS.

## 6. Conclusions

The application of ML technique to lung CT scan image processing can represent a valid tool to provide a broader adoption of CT scan quantitative analysis in the clinical practice of ARDS management, in particular the prediction of the alveolar recruitment and patient outcomes. Similarly, the application of AI to LUS imaging may implement clinician performance in distinguishing and interpreting similar LUS patterns deriving from different pathological etiologies with the potential to provide an accurate diagnosis (Figure 2). Indeed, there are several areas that could benefit from the application of AI in this field, including diagnosis, assessment of severity, progression, and response to treatment. However, AI is not ready for widespread use and models may not be as accurate because progression to advanced respiratory failure is not as common and predictable. In fact, part of the ML-based algorithms described in this review were based on image datasets collected during the COVID-19 pandemic. Nonetheless, updated algorithms have already been defined thanks to the ability to re-train those same algorithms with new image datasets. In this sense, AI is an evolving technology and an ongoing process of refinement and several biases should be overcome in the development of further models to guarantee sufficient robustness and reproducibility to be competitive compared with current standard methods and thus to support clinical judgment, starting from high quality imaging datasets.

## Figures and Tables

**Figure 1 jcm-13-00305-f001:**
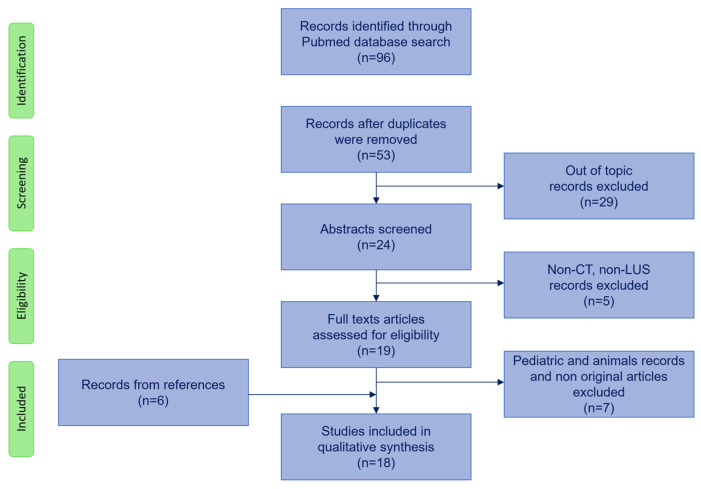
Flowchart of search strategy. CT: computed tomography; LUS: lung ultrasound.

**Figure 2 jcm-13-00305-f002:**
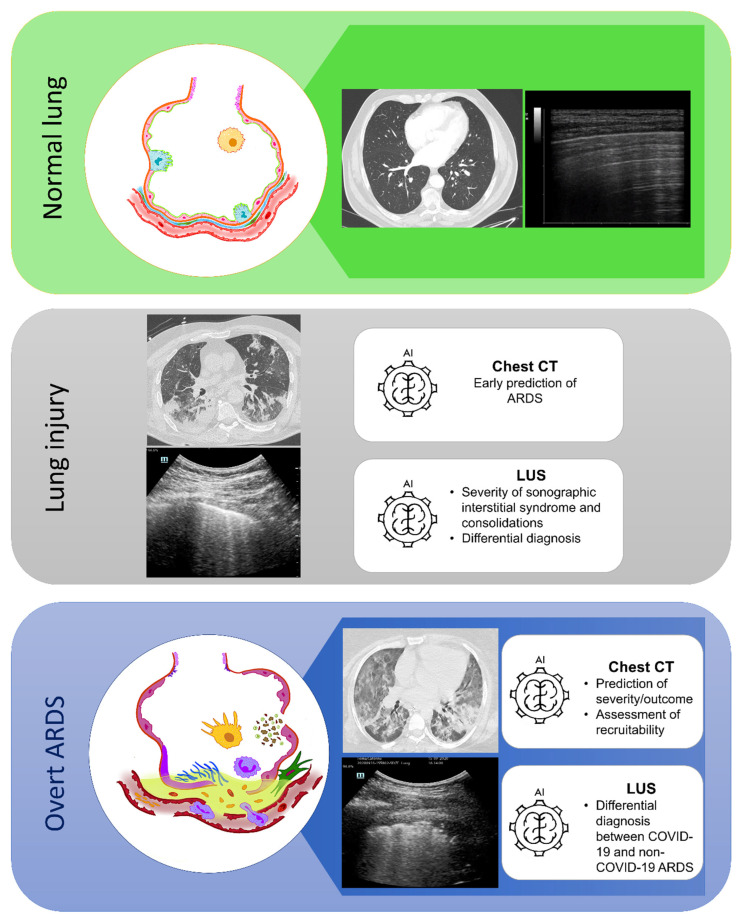
Current potential areas of application for artificial intelligence applied to lung computed tomography and lung ultrasound imaging in different stages of lung disease. Green upper box: normal lung histology (drawing **on the left**), axial projection of a lung CT scan (image **in the middle**), and a LUS scan showing normal pleural findings with repetitive physiological horizontal artifacts (A lines), a typical sign indicating a normal aerated lung. Grey box in the middle: a lung CT scan (**upper image**) and LUS scan (**lower image**) in a patient with acute lung injury. Note the presence of vertical artifacts arising from the pleural line (B lines), indicating the presence of a sonographic interstitial syndrome. Blue box at the bottom: overt ARDS (drawing on the left) with alveolar–capillary damage, alveolar edema, cellular debris, neutrophilic migration (in violet), activated macrophages (in yellow), fibroblast activation, and fibrin deposition (in green). The lung CT scan (**upper figure in the middle**) and the LUS scan (**lower figure in the middle**) represent the typical radiological findings in a representative patient with ARDS. Note the inhomogeneity of aerated and not aerated parenchyma at the axial projection of the lung CT and the irregular pleural profile, with areas of high lung density (white lung) interspersed by parenchymal subpleural infiltrates. The two different imaging approaches carry different qualitative and quantitative information of the same pathological pattern. AI: artificial intelligence; ARDS: acute respiratory distress syndrome; CT: computed tomography; LUS: lung ultrasound.

**Table 1 jcm-13-00305-t001:** Summary of the studies.

Author, Year	Study Design	Aim	Endpoints	AI Model	Results
Nishiyama, 2020 [9]	single center retrospective	prediction of prognosis	To evaluate the relationship between CT volume of well-aerated lung region and prognosis in ARDS patients.	An automated lung volumetry software of lung CT scan to identify lung region volumes by CT attenuation densities.	Well-aerated lung regions showed a positive correlation with 28-day survival. Survival outcome was better for percentage of well-aerated lung region/predicted total lung capacity ≥40% than <40%.
Gresser, 2021 [10]	single center retrospective	prediction of prognosis	To assess the potential of AI-based CT assessment and clinical score to predict the need for ECMO therapy in COVID-19 ARDS.	CT software provides segmentation of lung lobes providing a CT severity score.	AI-based assessment of lung involvement on CT scans at hospital admission and the SOFA scoring, especially if combined, can be used as risk stratification tools for subsequent ECMO requirement.
Hermann, 2021 [11]	multicenter retrospective	alveolar recruitment	To compare the accuracy in the computation of recruitability on CT scan between automatic lung segmentation performed by a properly trained neural network and manual segmentation in ARDS and COVID-19.	A DL algorithm to automatically segment ARDS injured lungs to calculate the lung recruitment.	The AI segmentation showed the same degree of inaccuracy of the manual segmentation. The recruitability measured with manual and AI segmentation had a bias of +0.3% and −0.5% expressed as change in well-aerated tissue fraction.
Kang, 2021 [12]	single center retrospective	differential diagnosis	To train a DL classifier model to differentiate between COVID-19 and bacterial pneumonia based on automatic segmentation of lung and lesion regions.	A DL model with deformable convolution neural network architecture trained to differentiate lesion patches of COVID-19 from those of bacterial pneumonia on CT scan.	DL lung CT scan analysis with constructed lesion clusters achieved an accuracy of 91.2% for classifying COVID-19 and bacterial pneumonia patients.
Lanza, 2020 [13]	single center retrospective	prediction of prognosis	To test quantitative CT analysis using a semi-automated method as an outcome predictor in terms of need for oxygen support or intubation in COVID-19.	Quantitative CT analysis with a semi-automated segmentation algorithm that divides lungs into not aerated, poorly aerated, normally aerated and hyperinflated.	The amount of compromised lung volume can predict the need for oxygenation support (between 6–23% of compromised lung) and intubation (above 23%) and is a significant risk factor for in-hospital death.
Liu, 2020 [14]	single center retrospective	prediction of prognosis	To quantify pneumonia lesions by CT (% of ground-glass, semi-consolidation and consolidation volume) in the early days to predict progression to severe illness using AI algorithms in COVID-19.	CT quantitative analysis combines a fully convolutional network with adopting thresholding and morphological operations for segmentation of lung and pneumonia lesions.	CT features on day 0 and 4, and their changes from day 0 to day 4, showed predictive capability for severe illness within a 28-day follow up. CT quantification of pneumonia lesions can early and non-invasively predict the progression to severe illness.
Pennati, 2023 [2]	single center retrospective	alveolar recruitment	To develop and validate classifier models to identify patients with a high percentage of potentially recruitable lung from readily available clinical data and from a single CT scan quantitative analysis at ICU admission.	Four ML algorithms (Logistic regression, Support Vector Machine, Random Forest, XGboost) to predict lung recruitment starting from a single CT scan obtained at 5 cm H_2_O at ICU admission.	The use of the four ML algorithms based on a CT scan at 5 cm H_2_O were able to classify lung recruiter patients with similar AUC as the ML algorithm, based on the combination of lung mechanics, gas exchange and CT data.
Penarrubia, 2023 [15]	single center retrospective	alveolar recruitment	To assess both intra- and inter-observer smallest real difference exceeding measurement error of recruitment using both human and ML on low-dose CT scans acquired at 5 and 15 cm H_2_O of PEEP in ARDS.	ML lung segmentation algorithm on CT scan to compute alveolar recruitment at 5 and 15 cm H_2_O of PEEP.	Human–machine and human–human inter-observer measurement errors were similar, suggesting that ML segmentation algorithms are valid alternative to humans for quantifying alveolar recruitment on CT.
Lopes, 2021 [16]	multicenter retrospective (study protocol)	prediction of prognosis	To develop a ML based on clinical, radiological and epidemiological data to predict the severity prognosis (ICU admission, intubation) in COVID-19.	A ML model receives a lung CT as input and outputs the stratification of lung parenchyma, discerning regions of the lungs with different densities.	Study in progress
Puhr-Westerheide, 2022 [17]	single center retrospective	diagnosis	To compare AI-based quantitative CT severity score to SOFA score in predicting in-hospital mortality at ICU admission in COVID-19 ARDS patients.	AI-based lung injury assessment on CT scan for the diagnostic performance to predict in-hospital mortality.	CT severity score was not associated to in-hospital mortality prediction, whereas the SOFA score showed a significant association.
Röhrich, 2021 [18]	single center prospective	prediction of prognosis	To develop a ML model for the early ARDS prediction from the first CT scan of trauma patients at hospital admission.	A ML model with convolutional neural network (radiomics) approach to automatically delineate the lung at lung CT to predict future ARDS.	The ML model with radiomics score resulted in a higher AUC (0.79) compared to injury severity score (0.66) and abbreviated injury score of the thorax (0.68) in prediction of ARDS. The radiomics score achieved a sensitivity and a specificity of 0.80 and 0.76.
Sarkar, 2023 [19]	single center retrospective	diagnosis and prediction of prognosis	To train and validate DL models to quantify pulmonary contusion as a percentage of total lung volume and assess the relationship between automated Lung Contusion Index and relevant clinical outcomes (ICU LoS and mechanical ventilation time).	DL model for automated CT scan segmentation to quantify the percent lung involvement indexed to total lung volumes.	Automated Lung Contusion Index was associated with ARDS, longer ICU LoS and longer mechanical ventilation time. Automated Lung Contusion Index and clinical variables predicted ARDS with an AUC of 0.70, while automated Lung Contusion Index alone predicted ARDS with an AUC of 0.68.
Wang, 2020 [20]	retrospective study	diagnosis	To explore the relationship between the quantitative analysis results and the ARDS existence, using an automatic quantitative analysis model based on DL segmentation model in COVID-19.	DL model to provide an automatic quantitative analysis of infection regions on lung CT to assess their density and location.	The total volume and density of the lung infectious regions were not related to ARDS. The proportion of lesion density was associated with increased risk of ARDS in COVID-19.
Zhang, 2020 [21]	single center retrospective	diagnosis	To compare the performance of the three DL models and determine which model is more diagnostic.	Three DL models (VGG, Resnet and EfficientNet) are used to classify LUS images of pneumonia according to different clinical stages based on a self-made image dataset.	EfficientNet showed to be the best model providing the best accuracy for 3 and 4 clinical stages of pneumonia, with an accuracy of 94.62% and 91.18%, respectively. The best classification accuracy of 8 clinical features of pneumonia at LUS images was 82.75%.
Baloescu, 2020 [22]	single center retrospective	diagnosis	To test the DL algorithm to quantify the assessment of B lines in LUS images from a database of patients presenting at ED with dyspnea or chest pain and to compare the algorithm to expert human interpretation.	A DL model is trained and developed based on a dataset of LUS clips to assess presence/absence of B lines and severity classification.	The accuracy in detecting B lines was 94% with a kappa of 0.88; the accuracy of the severity assessment was 56% with a kappa of 0.65.
Born, 2021 [23]	multicenter retrospective	differential diagnosis	To compare different AI models for the differential diagnosis of COVID-19 pneumonia and bacterial pneumonia.	Five AI models (VGG, VGG-CAM, NASNetMobile, VGG-segement, Segment-Enc) are tested on a dataset of LUS images and videos of healthy controls and patients affected by COVID-19 and bacterial pneumonia and compared in terms of recall, precision, specificity and F1 scores.	Two models (VGG and VGG-CAMI) had an accuracy of 88 ± 5% in distinguishing COVID-19 pneumonia and bacterial pneumonia.
Arntfield, 2021 [24]	multicenter retrospective	differential diagnosis	To compare the DL model and the surveyed LUS-competent physicians in the ability of discriminating pathological LUS imaging	A DL convolutional neural network model is trained on LUS images with B lines to discriminate between COVID-19 ARDS, non-COVID ARDS and hydrostatic pulmonary edema and compared with surveyed LUS-competent physicians.	The DL model showed an ability to discriminate between COVID-19 ARDS (AUC 1.0), non-COVID ARDS (AUC 0.934) and pulmonary edema (AUC 1.0) better than physician ability (AUCs 0.697, 0.704, 0.967).
Ebadi, 2021 [25]	multicenter retrospective	differential diagnosis	To compare the DL classifier model against ground truth classification provided by expert radiologists and clinicians.	A DL method based on the Kinetics-I3D network. classifies an entire LUS scan, without the use of pre-processing or a frame-by-frame analysis, for automatic detection of ARDS features present in pneumonia and COVID-19 patients (A lines, B lines, consolidation and pleural effusion).	The DL model showed an accuracy of 90% and a precision score of 95% with the use of 5-fold cross validation.

AI: artificial intelligence; ARDS: acute respiratory distress syndrome; AUC: area under the curve; CT: computed tomography; DL: deep learning; ECMO: extracorporeal membrane oxygenation; ED: emergency department; ICU: intensive care unit; LoS: length of stay; LUS: lung ultrasound; ML: machine learning; PEEP: positive end-expiratory pressure; SOFA: Sequential Organ Failure Assessment.

## Data Availability

Not applicable.

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
