# Peer review of "Lung Imaging and Artificial Intelligence in ARDS"

_jcm, 2024, doi:10.3390/jcm13020305_

Round 1
Reviewer 1 Report
Comments and Suggestions for Authors
Please see attached word document

Comments on the Quality of English LanguagePlease see attached word document. Multiple sections need to be rewritten.
Reviewer 2 Report
Comments and Suggestions for Authors
In this manuscript, the authors summarized the state of art of AI in the field of lung imaging, focusing on CT and ultrasound technique in ARDS patients. On this basis, they discussed the advantage of AI in assessing the prediction of ARDS, the quantification of alveolar recruitment, the possible alternative diagnosis and the outcome. This work emphasized that the application of AI to lung CT scan image processing and LUS imaging could assist clinician to predict and diagnose ARDS, quantify alveolar recruitment, differentiate different pathological etiologies of ARDS and predict the outcome. The manuscript is well-organized and clearly stated.
1. In order to guarantee the comprehensiveness of the this review, it may be necessary to retrieve multiple databases rather than a single database.
Comments on the Quality of English LanguageNo
Reviewer 3 Report
Comments and Suggestions for Authors
Review report: November 18, 2023
The review manuscript entitled” Lung Imaging and Artificial Intelligence in ARDS” is well designed and prepared. During the review process, some minor and major critiques were raised and need to be addressed to improve the quality of the manuscript and be fit to JCM.
Minor critiques:
1- Misspelling of intelligence in the title of article should be corrected.
2- The structure of abstract should be based on what the authors read and concluded in new phrasing and expression, not repetition of quoted sentences that already used in the other body parts of the manuscript. Please rewrite the abstract.
3- Reference #24 does not have a year of publication.
4- In figure 2, the top panel need to explanation in its legend. E.g. what does the black box represent?
5- Line 164 -166: Did the mortality decreased due to early diagnosis with AI-based tools or due to improved treatment protocols?
Major critiques:
1- The modernity of quoted references is good, 33 out of 55 between 2019 and 2023. However, the number of cited references is still low and insufficient to cover all the controversary hypothesis.
2- All systems, tools and techniques used to diagnosis/prediction of ARDS should be briefly listed including X-ray, biopsy, smearing …etc and how they assisted in this matter before and after AI revolution.
3- A comparison of recent techniques that depend on AI with old technology used to be used in ARDS diagnosis\prediction should be included.
4- Graphical abstract could improve the quality of this review. Remember to include the old techniques too in this graphic.
5- Conclusion section should present what the authors could conclude from using AI in diagnosis or prediction of ARDS in comparison with previous tools.
Comments on the Quality of English LanguageThe title has misspelling of intelligence.
Round 2
Reviewer 3 Report
Comments and Suggestions for Authors
Thanks for addressing our concerns in point-by-point basis.
Author Response
We thank the reviewer for taking the time to review this manuscript.